# Attachment, Shame, and Trauma

**DOI:** 10.3390/brainsci15040415

**Published:** 2025-04-19

**Authors:** Carol George

**Affiliations:** Department of Psychology, Mills College at Northeastern University, Oakland, CA 94613, USA; cygeorge@comcast.net

**Keywords:** attachment, shame, interpersonal neurobiology, attachment trauma, adult attachment projective system

## Abstract

Background/Objectives: Early parent–child attachment interactions guided by right-to-right brain synchrony are the foundation of emotional development and the quality of attachment relationships. Interactive failures are the hallmark of not only insecurity and trauma but also the internalization of shame. The purpose of this study was to advance our understanding of the relation between attachment and shame. The study explored a range of shame experiences, from normal socialization to harsh treatment and abuse. Debilitating shame was expected for individuals who had not mourned attachment trauma. Methods: Attachment was assessed using The Adult Attachment Projective (AAP) system in a sample of 245 adults. The AAP identifies the traditional regulated attachment classifications (Secure, Dismissing, Preoccupied) and, in addition to Unresolved attachment, three forms of incomplete mourning—Failed Mourning, Preoccupied with Personal Suffering, and Traumatized Secure. The study used participants’ narratives regarding three AAP classifications when “alone” to examine individual differences in representations of the shamed self contexts portraying the self in Private, Exposed, and Threatening situations. Results: All study hypotheses were confirmed. Adults with regulated attachments (Secure, Dismissing, Preoccupied) were significantly less likely to ascribe shame to the AAP pictures than traumatized adults. The patterns of results were the same when comparing differences in shame intensity and outcomes. There were no differences in shame intensity in the regulated group. Shame intensity for the Traumatized Secures was somewhat greater. There was a significant increase observed in the traumatized groups, with the Unresolved group showing the highest ratings. Reparative outcomes were significantly related to attachment security, although not to the extent expected. Secure, regulated insecure (Dismissing, Preoccupied), and some traumatized individuals described reconciliation and functional restitution. Unresolved individuals left shame unremedied. Regression analysis demonstrated that attachment classification was a greater predictor of shame intensity than outcomes. Conculusions: Attachment and neurological development are intertwined. Childhood interactive failures are deeply traumatic. If not mourned, shame takes its place in the identity core. This study provides clinicians with an in-depth perspective on attachment and shame assessment for goal-setting in therapy, consonant with their patients’ attachment representations.

## 1. Introduction

Shame is a painful feeling of humiliation or distress caused by conscious awareness of wrong or absurd behavior [1]. Often, the words used to identify shame are embarrassment, humiliation, disgrace, chagrin, and loss of face. “Shame is the affect of indignity, of defeat, of transgression and of alienation… shame strikes deepest into the heart of man. [He] feels shame is felt as an inner torment, a sickness of the soul…He feels himself naked, defeated, alienated, and lacking in dignity or worth” [2].

The origins of shame are rooted in the study of emotional development, attributed to the early interpersonal interactions of a mother and her infant, in particular right-to-right brain synchrony. Early shame shapes the individual’s views of self and relationships throughout the life span. Attachment defines the early interpersonal and emotional bond between parent and infant, which has lifelong implications for physical and mental health. The clinical literature argues with little empirical evidence that shame is housed in insecure relationships. Although this thinking makes surface sense, to equate shame with attachment insecurity fails to acknowledge the nuances of attachment patterns, the meaning of insecurity, and the processes of the development of a shamed identity that functions as the developmental core. The purpose of this study is to refine this thinking by examining shame in the attachment mental representations of adults.

The emotion theorist Silvan Tomkins [3] described observations of early shame expressions in infancy around seven months. Self-awareness emerges in the second year of life along with mental representations for knowing how to relate to others [4]. When the self is “exposed” to others, their emotional experiences are no longer private; self-conscious emotions such as guilt, shame, jealousy, and pride join the emotional repertoire [5,6]. In a study of infant emotions, Lewis and his colleagues [6] reported that toddlers’ first reaction to seeing themselves in a mirror was embarrassment, first visually checking in with their parent and then engaging in classic shame behavior, including averting their gaze and touching their hair, hands, or face. Erikson’s [7] lifespan psychoanalytic developmental framework poises parents as pivotal in supporting the toddler in finding a balance between autonomy (i.e., I can do it myself) and becoming ashamed and doubting their competence.

The developmental timeline of shame coincides with the ages children are consolidating attachment relationships with primary caregivers [8]. This developmental convergence in infancy and toddlerhood suggests shame is likely intertwined the quality of the child’s attachment to their parents, yet there is scant attention to shame in the attachment literature. H.B. Lewis [9] and others [10,11] called shame an attachment emotion embedded in the nervous system like attachment.

### 1.1. The Neurobiology of Attachment

Attachment theory provides a powerful framework for understanding the biological foundations of shame. Attachment theory is an evolutionary-based theory of a specific type of human social relationship that begins in infancy and has developmental consequences across the lifespan. Attachment is a biologically based behavioral system that evolved in ways that influence motivation, cognition, emotional regulation, and memory. Attachment represents an organized set of behaviors and neurophysiological networks that provide protection and care [12,13]. The attachment system is thought to be an organizing feature of basic neurological function and the brain [14]. A review of the neuro- and biological correlates and sequelae of attachment in children and adults is beyond the scope of this paper. Some discussion is needed to set the stage for this study and the central role attachment plays in the development of shame.

The reciprocal relationship between the infant and caregiver is central to the development of attachment. Looking at the parent contribution, research shows, for example, that mothers exhibit bilateral orbitofrontal cortex activation when they see their own babies as compared to unfamiliar babies [15]. Recent research shows maternal representations of their childhood attachment experiences are related to children’s amygdala and hippocampal volumes. Maternal security was associated with children’s smaller left amygdala volume and larger surface area in the right cortex [16].

Research confirms biological regulatory differences related to adolescent and adult attachment patterns. Buchheim and her colleagues have carried out the most comprehensive work in this area. In one study, adolescents with Secure attachment assessed using the same measure as used in the current study showed greater heart rate variability under stress than Dismissing and Unresolved adolescents [17]. When compared to the control neutral pictures, attachment scenes activated the inferior parietal lobes (IPLs), the middle temporal gyrus (MTG), and the anterior medial prefrontal cortex (mPFC). These areas are associated with reasoning about mental representations, semantic memory of social knowledge, and social cognition [18]. One of the most notable results from this study was evidence of significantly greater activation of the right temporo-parietal region in response to attachment scenes depicting dyads (i.e., two people interacting together). This finding is central to arguments for the importance of parent–child right brain attunement.

### 1.2. The Biological Intersection of Attachment and Shame

Examining its evolutionary history shows that shame is rooted in submissive appeasement behaviors used to regulate social rank and status [10,11]. Darwin [19] wrote that social animals were “under an obligation” not to mortally harm each other. This is because shame behavior emerges in interactions involving interpersonal power or transgressions [11]. The weaker individual must demonstrate their diminutive status, including efforts to become small, not seen by others, or invisible—downcast eyes, turning away, crouching down, or hiding their face. Evolutionary biologists explain these apologetic appeasement behaviors as efforts to avoid life threats, including being expelled from the group [11]. These behaviors are efforts to keep the relationship intact by reducing the threat to repair, reconcile, or manage the relationship [11].

Schore’s [13] model of interpersonal biology provides a comprehensive discussion of the neurophysiology of shame. His thinking is central to the understanding of the development and management of emotional development in “heightened affective moments” such as shame in the early mother–child attachment relationship. Consistent with Tronick’s [20] descriptions of parent–child en face interaction, Schore describes how the dyad creates a synchronized regulatory ballet of mutual attunement–engagement, disengagement to recover from heightened emotional intensity, and re-engagement. Like attachment, the synchrony between parent and child is biologically primed in the brain with channels of communication mediated by eye-to-eye contact and behavioral expressions (e.g., hand gestures) of interpersonal interest. Schore proposes that, as the infant matures, these interactions create the “final circuit wiring” in the experience-dependent right orbitofrontal cortex, the area of the brain related to attachment social interaction. This portion of the brain is exceptional because it is linked to limbic areas.

Drawing from Tomkins [3], Schore views shame as an emotional inhibitor that reduces or blocks enjoyment, interest, and exploration early in the second year of life. The parent’s failed attunement interrupts the infant’s expected regulatory interactions and ability to restore emotional homeostasis. Tronick [20] shows using a still-face paradigm how infants attempt, sometimes in desperation, to reconnect visual engagement with a stone-faced parent who refuses even for a few moments to engage.

Schore explains how shame results from experiences of ending emotional contact and “precipitates a rapid and unexpected contraction of the self”. His descriptions of shame give neurophysiological meaning to Darwin’s behavioral descriptions. The infant recovers only if the parent re-engages to repair their distress. Shame, stress, and reparation are ubiquitous occurrences during infancy and are essential for right orbitofrontal cortex development. These interactions contribute to the emotional experiences of shame in the infant’s early attachment representations and support their agency to create emotion-related environments that continue into adulthood.

Schore defines the neurophysiological regulatory breakdown as trauma. The infant recoils and experiences the self as an ineffective contributor to the relationship. The need for attachment is genetically embedded in the primate nervous system [13]. In attachment theory, the kind of breakdown Schore describes is a collapse in the attachment–caregiving relationship termed “disorganization” based on observations of the infant upon reunion with the parent following a separation [21]. Solmon and George [22] suggest that even if the parent fails to meet the infant’s biologically based expectation, a collapse and neurophysiologically “programmed” attempt for a reunion with the attachment figure are impossible. The infant loses accessibility to the attachment figure, even if the attachment figure is physically present. For the purpose of this study, shame, like loss through death, is conceptualized as a separation and potential loss of the attachment figure. According to Bowlby [23], reunion and repair are required to prevent feeling frightened, isolated, and abandoned. Thus, the only way to repair shame is to mourn.

### 1.3. A Developmental Attachment Approach to Shame

Clinical psychologists have historically dominated the discussion of shame. Attention to the origins and mental health consequences of shame is often attributed to Helen Block Lewis [9]. More recently, clinicians have pointed to the strong connection between shame and interpersonal trauma, e.g., [24,25,26,27,28,29], a connection that naturally leads to questions about the contributions of parents to their clients’ shame.

Shame is uncomfortable, and people go to great lengths not to use the word shame or ascribe feeling ashamed. In reality, shaming in human child–parent relationships is ubiquitous, which suggests a normative function for shame. Developmentalists understand behavior as being on a continuum where aberrations are extreme forms of normative developmental pathways. They suggest that in its normative form, parental shaming is healthy because it serves as a protective function by sending the child a strong message of behavioral inhibition for undesirable behavior. Normative shaming reduces the risk of future harm and family or community ostracization [11].

Clinical psychologists typically see patients whose parents failed to protect them, often because they are the source of physical or psychological harm. The intensity of their clients’ experiences led to developing terms for shame that from a developmental perspective are unnecessarily harsh, for example “terrorizing”, “toxic”, or “chronic” shame, e.g., [28,30,31]. The problem is that they ascribe shame to experiences of parental rejection. Attachment theory and research show a mismatch between how clinicians and attachment theorists define rejection.

In attachment, rejection encompasses parents’ efforts to mute or sideline the child’s emotional and physical needs by downplaying their importance [29]. It compromises security but is not the kind of pervasive threat to relationship integrity. Rejection is a normative defense against uncomfortable emotions and relationship intimacy [30,31]. Children respond to parental rejection in ways ethologists call conditional attachment strategies, ways that help children stay close enough to their attachment figures to be protected [31].

Rejection defines the distanced emotional and behavioral interactions of parents and children described as insecure–avoidant or dismissing adults [8,32]. Despite the emotional cost, these parents and children are confident their relationship is working, and parents will attend to their children’s attachment needs if necessary. The resulting emotional and interactive distance generates a sense of normality without having to see the compromises to security.

In understanding shame, we must differentiate “good enough” insecure relationships from risk and not assume that all rejection is toxic or chronic. Parental rejection is likely accompanied by some degree of shame because it temporarily blocks the child’s direct access to their attachment figure. Normative shaming sends the message, “Stop in your tracks. I need you to pay attention to changing your behavior so I can protect you”.

What clinical psychologists are describing is unhealthy shame. Shame becomes a destroyer that threatens to shatter the attachment–caregiving bond and the developing self. The message is “Stop in your tracks, or I’ll abandon and not protect you anymore”. This message bodes for relationship collapse and tells the child they are in danger.

Feeling endangered is the essence of attachment dysregulation and trauma [29,30,32]. The child interprets the parent’s failure as abdicating their biological role as protector. The child assumes they must not be good enough in the parent’s eyes to receive their care. Repeated experiences of abdication create internalized shame because they foster an enduring sense of self as underserving and worthless.

In attachment theory, parent abdication is the core of attachment disorganization [29]. The accompanying pain, fear, anxiety, and feelings of isolation and helplessness dysregulate attachment at the behavioral and neurophysiological level [22,30,32]. The overlap between these dysregulated attachment affects and those described by shame scholars is not coincidental e.g., [5,9,27,28].

The most effective mechanism to restore emotion regulation is to repair it with an apology—originating either from the child or the parent [11]. Kaufman [27] points out that restoration can be complex because shame is “rarely a wholly conscious process” and, if conscious, is not spoken. Restoration then appears to be easier said than done.

Tomkins [3] also views shame as a loss, just like loss through death is a loss. As previously stated, loss must be mourned. George and West [30] expanded Bowlby’s ideas regarding mourning to apply them to all forms of attachment trauma. Some never complete it. Bowlby [23] applied the term “pathological mourning” to describe incomplete grief. Some individuals fail to mourn (i.e., they never begin), while others are consumed by grieving, becoming preoccupied with or unpredictably flooded by their suffering. Pathological mourning is associated with a range of maladaptive defenses that overlap with clinically described shame defenses. Traumatically shamed individuals become enraged, blame others for their problems, express envy and jealousy, aspire to perfectionism, or withdraw into themselves (including dissociation) [24,27].

There are only two published attachment studies on shame. Both evaluated the parent’s representations of the shamed self using narratives about hypothetical shame contexts drawing from picture stimuli in the Adult Attachment Projective Picture (AAP) System [30]. The AAP is an empirically validated free response measure of adults’ mental representations of childhood attachment experiences. The measure asks individuals to tell a story about the events and feelings related to seven attachment-related pictures. Solomon and George [11] were interested in the patterns of mother’s representations of the shamed self in parent–child conflict situations and the relation of their shame patterns to their children’s attachment. They chose the parent–child conflict because early shaming most frequently occurs in the family. The AAP picture that best “activates” stories of parent–child conflict depicts a standing alone in a corner.

Solomon and George evaluated shame from narrative descriptions of interactions and negative attributions of the child to examine the relation between maternal representations of the shamed self and their children’s attachment patterns. Ninety percent of the mother’s responses to this picture described some form of parental shaming. Forty percent of the stories described normative shame (e.g., the child breaks a house rule). Most stories depicted harsh shame, where the child was viewed as the perpetrator and the parents’ response was extraordinarily punitive or frightening. Integrative repair (e.g., an apology) was more common for mothers of secure than insecure children compared with the story outcomes for mothers of insecure children that were mostly incomplete or unremedied.

Engberg Conrad and her colleagues [33] investigated adult attachment representations of shame in a sample of parents of autistic children also using the AAP. In addition to the image of the cornered child, they examined responses to a picture of a lone person sitting on a bench. The central finding of this study was the outcomes of the stories of Unresolved parents who described incomplete or unremedied shame.

These two studies demonstrate how parents internalized representations of the shamed self that are embedded in attachment. These researchers were surprised by the number of stories depicting overly tense or harsh interactions. The combined study results showed the ubiquitous nature of parental shaming, the rarity of repair and reconciliation outcomes, and the central contribution of dysregulated attachment (disorganized children, Unresolved adults) to the internalization of deep traumatizing shame.

### 1.4. The Current Study

The purpose of the current study is to extend these two studies by elaborating on the developmental contributions of attachment trauma to representations of the shamed self. Schore’s [13] detailed model of interpersonal neurobiology shows how mother–child right brain attunement breakdowns sows the seeds of traumatic shame. The idea of a relationship breakdown is consonant with attachment theory’s perspective that parental abdication and failure to assuage children’s fears threaten the attachment relationship and are traumatic at the neurophysiological, behavioral, and representational levels [22]. Rejection in attachment theory is not equivalent to a breakdown. It follows that attachment insecurity, although with costs, is regulatory, and it is attachment trauma that feeds toxic or chronic shame.

Using AAP narratives of the representational self, the current study evaluates attachment group differences in adult representations of internalized shame. Following Engberg-Conrad et al. [33] the basic comparison groups are regulated (Secure, Dismissing, Preoccupied) versus traumatized attachment (pathological mourning groups and traumatized Secure). Bowlby [12] proposed that the most excruciating human experience is to be alone because it threatens access to protective and comforting attachment figures. In that vein, the current study will evaluate the representational intensity of the shamed self using three AAP pictures: the Private self (Window), the Exposed self (Bench), and the Threatened self (Corner).

The current study hypothesizes differences between regulated and traumatized patterns in the

(1)Frequency of shame stories in response to AAP pictures depicting the Private, Exposed, and Threatened self;(2)Intensity of shame revealed in response to the Private, Exposed, and Threatened self;(3)Shame outcomes in the stories depicting the Private, Exposed, and Threatened self.

## 2. Materials and Methods

### 2.1. Participants

Data for the two combined cross-sectional attachment studies were conducted simultaneously in the same lab to investigate correlates and sequelae of attachment. Participants were recruited from communities in Northern California, USA. The Human Subjects Committee Internal Review Board of Mills College approved both studies. Approval ensures that participants gave signed consent to participate in the studies. One study was an investigation of mothers’ child abuse risk. The sample comprised 145 women with an average age of 37.54 years (SD = 6.94) with young children with an average age of 54.5 months (SD = 10.65, 55% girls). Participants were predominantly college-educated (*n* = 104, 71.7%). Fifty identified as white or multicultural (65.5%). The other study was an investigation of adult traumatic stress. The sample comprised 48 women with an average age of 19.88 years (SD = 12.44) and 52 men with an average age of 27.27 years (SD = 9.97). Ten women (35.7%) were college-educated, and 34 (70.8%) identified as white or multicultural. Twenty men (51.9%) were college-educated, and 34 identified as white or multi-cultural (65.4%). The sample of mothers was significantly older and more educated than the stress sample, (t243 = 7.80, *p* < 0.001, X1 = 29.1, *p* < 0.001, respectively), with no between-sample culture differences. There were no differences between women and men in age, education, or culture.

### 2.2. Measures

#### 2.2.1. Attachment Pattern Classification

Attachment classification patterns were assessed using the Adult Attachment Projective Picture (AAP) System [30]. The AAP comprises eight black-and-white line drawings. The scenes depict theoretically derived attachment events [15]. The drawings contain only enough detail to identify an event; strong facial expressions and other details are absent. The character depictions are drawn without culture, gender, or age biases. The scenes in the order of administration are as follows: Neutral—two children play with a ball, and seven attachment pictures beginning with Child at Window—a child looks out a window; Departure—an adult man and woman with suitcases stand facing each other; Bench—a youth sits alone on a bench; Bed—a child and woman sit opposite each other on the child’s bed; Ambulance—a woman and a child watch a stretcher being loaded into an ambulance; Cemetery—a man stands at a headstone; and Child in Corner—a child stands askance in a corner. Some scenes portray the main character as alone (“alone” pictures). Others portray adult–child or adult–adult dyads (“dyadic” pictures) intended to depict an attachment-caregiving relationship.

Studies using the AAP studies demonstrated significant psychometric properties of the AAP by showing high inter-rater reliability [30,34,35,36] and high concurrent validity with the AAI. Concordance rates for the four-group attachment classifications (F, Ds, E, U) were 90%, κ = 0.84, *p* < 0.001, and for the two group attachment classifications (secure vs. insecure), they were even 97%, κ = 0.89, *p* < 0.001 [30,36]. A recent study on adult patients with chronic depression and non-depressed controls demonstrated high convergent validity between the AAI and the AAP in *N* = 30 participants, κ = 0.89 (ASE = 0.112), *p* < 0.001, simple agreement 94% [37].

The AAP was administered individually in a private office. Pictures were presented in the order listed above using the same probes for every picture: what is happening in a scene, what led up to that scene, what the characters are thinking or feeling, and what happens in the end. Responses are audio-recorded and transcribed for analysis. Administration is typically 25 min.

The classification pattern is determined by analyzing the verbatim transcript of the narrative responses, called stories, to the attachment scenes (i.e., a neutral story is not included). They are evaluated for (1) narrative, (2) content, and (3) defensive processes. The judge evaluates these dimensions separately for each story and decides on a classification based on the overall story patterns. The narrative dimension evaluates the stories for elements of personal experience. The AAP is not an autobiographic assessment; it does not ask about real-life situations or experiences. The inclusion of personal experience material indicates difficulties maintaining self–other boundaries and is often seen in the stories of individuals who are judged as Preoccupied or Unresolved. The story content dimensions evaluate how the narrative conveys meaning to the relationships depicted in the storyline. Two dimensions are evaluated for the stories about aloneness. One is the Agency of Self, which reveals the degree to which the interviewee can envision the projected self engaging in constructive action or thoughtful exploration regarding the self or the story relationships. The other is Connectedness, which shows the interviewee’s desire and capacity to be connected in intimate relationships with parents, friends, or romantic partners. The dyadic stories are evaluated for Synchrony. This dimension demonstrates variations in the interviewee’s representation of caregiver sensitivity or mutual enjoyment.

The AAP is the only adult attachment assessment that shows how individuals use defenses to regulate emotional arousal. Defensive processes transform thoughts and emotions to buffer the individual as much as possible from discomfort or dysregulation. Defenses are evaluated for all seven attachment stories by examining the narrative’s words, images, and patterns. Attachment theory defines three forms of defense: deactivation, disconnection, and segregated systems [18]. *Deactivation* defenses cool down emotional discomfort by minimizing emotional intensity (e.g., “all children experience some kind of bullying at school”, “always follow social or parents’ rules”) or shifting attention away from attachment themes to tangential developmental themes such as achievement and competence, job-related tasks, or interactions with peers instead of parents. Deactivation limits the kind and degree of emotions expressed (e.g., characters are sad, never depressed); the message is that emotions, especially anger, are not part of the conversation. *Disconnecting* defenses split attachment information and affect from the situations or people involved. Disconnection undermines the individual’s ability to “see” and describe a unitary, consistent story and evidences a fractured attachment state of mind. Stories are vague or do not describe a single theme. The characters are in limbo, waiting, wondering, wishing, or hoping for something to happen. Emotions, positive and negative, can dominate the conversation. The emotional goal of disconnection is to be happy, and managing negative affect is central to the path to happiness. The stories reveal a mental preoccupation with anger, anxiety, and frustration, including attempts to withdraw from situations or people who are the source of emotional distress to achieve this goal.

Deactivation and disconnection are conceived as “normative” regulating defenses [32]. Both forms help individuals regulate attachment distress. The third form of defense evidences representational dysregulation. Bowlby [23] called this form *segregated systems*, developed to modernize the concept of psychoanalytic repression. It is an extreme form of defensive exclusion produced by experiences of unremedied loss. George and West [30] expanded Bowlby’s concept to apply, not only to loss but also to all chronic or severe attachment threats involving attachment figure protection failures. These experiences and the associated affect are “packaged up” with the attempt to lock them away (literally segregate) from consciousness [23]. The AAP is stressful and uncovers segregated trauma that risks emotional neurophysiological and emotional dysregulation [38]. Normative defenses that monitor and regulate attachment experience and emotional break down. Segregated trauma themes include descriptions in the story of, for example, fear, helplessness, threats by others (including attachment figures), pain, isolation, or abandonment. In some stories, segregated systems emerge in descriptions of dangerous themes (e.g., jumping out of a three-story window), or the extreme anger of enraged parents. Some themes include images that are interpreted using studies of disorganized children or adults unresolved by loss, such as dissociation, role reversal, and compulsive caregiving for others [23,29,39].

The defining qualities of Secure adult attachment is relationship satisfaction, conflict repair, and flexible emotional integration. The secure–flexible AAP story pattern provides a rich examination of the past that has created a representation of self as worthy of attachment-figure comfort and protection and trust that parents and other attachment figures will respond in kind if needed. Stories describe self and relationship bumps and imperfections remedied by integrated agency, the desire to be connected in intimate relationships, and reciprocal sensitivity, comfort, and mutual enjoyment in attachment and other relationships. The speaker distinguishes in their mind between what is hypothetical and personal, maintaining during storytelling the boundaries between self and other, past and present. Secure individuals use defenses sparingly to support homeostasis rather than exclude uncomfortable emotional elements in their stories. If trauma is present (i.e., evidence of segregated systems), comfort or thoughtful exploration regulates the traumatic affect.

The defining quality of Dismissing attachment is defensive deactivation as the primary coping strategy throughout the AAP. Deactivation not only maintains interpersonal boundaries, but it creates emotional and physical distance between the self and other story characters. Integrated themes of agency and synchrony are rarely depicted. The purpose of relationships is to provide basic care (e.g., a ride to school); the interactions lack intimacy, comfort, or mutual enjoyment. The story themes minimize, reject, or avoid negative affect and outcomes. If trauma is present in the story, functional action (e.g., the child says, “Go away”) or help from others regulate but do not assuage the distress; representations of homeostasis and rebalancing are rare.

The defining quality of Preoccupied attachment is defensive disconnection as the primary form of coping strategy throughout the AAP. Preoccupied individuals tell confusing, vague, and emotional stories. Boundaries between themes and hypothetical versus personal events are often murky. As with Dismissing attachment, integrated themes of agency and synchrony are rarely depicted. The lack of intimacy, comfort, or enjoyment is camouflaged by glossy statements of excessive optimism. The stories demonstrate how confused and worried Preoccupied individuals are about attachment events and the people associated with them. If trauma is present in the story, the projected self muddles through (e.g., the girl on the bench hopes someone will come and ask her how she is doing). Blurring out traumatic distress and combining it with the hope that situations will work out temporarily moderates negative affect.

Unresolved attachment can have elements similar to any of the regulated patterns—Secure, Dismissing, or Preoccupied. The term “unresolved” was coined by Main [21] to apply to Bowlby’s description of never-ending grief of the death of a loved one [23]. George and West [30] showed that the concept of being unresolved applies to all forms of traumatic losses, not just loss through death. The defining quality of the Unresolved classification is the evidence that the speaker becomes flooded by trauma and that regulating this invasion is impossible. There are two Unresolved AAP responses. One is a flooded and dysregulated narrative (i.e., trauma is not managed or contained); the other is a constricted response where the individual freezes up and cannot describe anything about the picture stimulus (i.e., an instinctive mammalian fear response). The Unresolved pattern often cannot control the intrusion of traumatic elements of their personal experience.

In addition to Unresolved attachment, the AAP identifies three other incomplete mourning patterns. The mourning is incomplete, but unlike Unresolved attachment, the speaker packages trauma and experiences of parental failed protection with regulating defenses [30]. Two forms were described by Bowlby [23]: Failure to Mourn (i.e., blocked mourning) and Preoccupied with Personal Suffering, associated, respectively, with hyperactivation of defensive deactivation and disconnection. George and West [30] also showed that not all secure individuals had fully completed mourning attachment trauma, identifying this group as “Traumatized Secure”. These three additional trauma groups are based on the intensity of segregated systems content in the AAP story narratives and empirically validated to identify debilitating trauma, e.g., [40].

In summary, the AAP identifies seven distinct attachment classification patterns. The Secure (F), Dismissing (Ds), and Preoccupied (E) are conceived as regulated (sometimes referred in the attachment literature as organized). Traumatized Secure (Ftr), Failed Mourning (Dstr), Preoccupied with Personal Suffering (Etr), and Unresolved (U) describe incomplete grief patterns and are conceived as dysregulated by lingering trauma.

#### 2.2.2. Shame

Mental representations of the shamed self were evaluated in AAP stories by examining the presence of shame themes and its outcome (i.e., how the shame was remedied). Feeling ashamed can be most intense when individuals are alone. Therefore, the study used the three “child” alone stimuli representing the Private (Window), Exposed (Bench), and Threatened (Corner) self.

Shame is identified based on three elements attributed to shame in the literature. One is the speaker naming shame or its equivalents in the story. The second is descriptions of physical behavior, defenses, or core internalized evaluations of the shamed self. The third is contexts or interactions conducive to shame. The shame categories in AAP stories are listed in Table 1.

Shame stories (i.e., where shame evidence was present) were evaluated for intensity and outcome. A 7-point scale was developed to evaluate shame intensity based on a qualitative assessment of shame indicators appearing in normative and harsh shame situations. See Table 2. Shame outcomes comprised six ordinal categories, ranging from integrative repair to unremedied shame. See Table 3.

Shame outcome was defined by relative integration and repair, with ratings assigned as follows. (1) Integration or repair: the story is coded with the AAP agency categories of internalized secure base or haven of safety. (2) Restoration: Themes of comfort in a close relationship that is not attachment (e.g., the girl’s friend comforts her) or parent instrument reconciliation that includes an explanation or lecture. (3) Functional: Themes of comfort by individuals not in a close relationship (e.g., teacher nurse) or the child approaching the parent or vice versa to obtain information for restitution (e.g., how much the window the child broke will cost to repair). (4) Self-manage: The main character takes care of their shame alone (e.g., the child is bullied at school and goes home). (5) Enduring: The narrative evidences that attempts at a restoration, functional, or self-management outcome do not work (e.g., the parent comforts the child and tells them how to do better in the future, but the child is still upset and goes to their room to hide). (6) Unregulated: There is no attempt to regulate the character’s shame (e.g., the parent screams and hits the child and then leaves the room; the child sits in the corner and cries).

The shame coding and rating scales were developed for the current study. Outcome coding was an extension of a rubric developed by Solomon and George [11]. Twenty percent of the sample was evaluated for reliability on these dimensions by the author and independent judges who were blind to all variables and hypotheses in the current study. There was significant agreement for the categorical variables shame and outcome: Kappas = 0.85 and 0.90, *p*’s < 0.001, respectively. Pearson correlation for shame intensity was 0.87, *p* < 0.001. Attachment classification reliability was Kappa = 0.88, *p* < 0.001.

### 2.3. Data Analysis and Tests for Covariates

There were no significant associations between the participant demographics and the dependent variables (shame category, shame intensity, outcome). A chi-square analysis showed that there were no sample differences in the distribution of attachment classification groups; therefore, the mother and adult stress samples were combined for analysis. The sample attachment distribution is 82 (33%) participants in the regulated attachment group (33 F, 31 Ds, and 18 E) and 163 (66%) participants in the trauma group (36 Ftr, 53 Dstr, 18 Etr, and 56 U). Chi-square was used to evaluate group differences. The analysis of shame intensity used nonparametric tests.

## 3. Results

### 3.1. Shame and the Alone Self

Hypothesis 1 predicted differences in the frequency of shame stories between regulated and trauma classification groups across the Private, Exposed, and Threatened shame settings. The results showed significant differences between the groups, χ^2^ = 13.84, *p* < 0.001, χ^2^ = 36.76, *p* < 0.001, and χ^2^ = 7.66, *p* < 0.01. The attachment classification pattern frequencies for shame stories are shown in Table 3. There were significant differences between the private and exposed self, with a significant trend level for the threatened self. Shame stories predominated in the Exposed and Threatened self settings and were predominant for all groups in settings portraying the Threatened self. There was little shame when the child was in a private setting. Shame behavior predominated in stories of the exposed self on the bench. Shame involving shame contexts predominated for the threatened child in Corner.

### 3.2. Attachment and Shame Intensity

Hypothesis 2 predicted differences in shame intensity in shame stories between the regulated and trauma attachment groups. This hypothesis was supported. The shame intensity ratings for the regulated group were significantly lower than those for the trauma group. Ratings for the Private self-regulated group (*n* = 8, M = 1.25, SD = 0.36) were significantly lower than the trauma group (*n* = 49, M = 3.26, SD = 1.73), Mann–Whitney U = 336.00, *p* < 0.001. Ratings for the Exposed self-regulated group (*n* = 41, M = 1.63, SD = 0.83) were significantly lower than the trauma group (*n* = 142, M = 3.13, SD 1.55), Mann–Whitney U = 4580.00, *p* < 0.001. Ratings for the regulated Threatened self (*n* = 61, M = 1.80, SD = 0.81) were significantly lower than the trauma group (*n* = 154, M = 3.87, SD = 1.52), Mann–Whitney U = 8062.00, *p* < 0.001.

The Kruskal–Wallace statistic was used to examine differences among the attachment groups. The results are shown in Table 4. There were no significant differences in shame intensity among the regulated attachment groups. There were significant differences among the trauma groups, however, showing significantly lower intensity ratings for the Ftr group than the other trauma groups, increasing for Dstr and highest for Etr and U.

### 3.3. Attachment and Shame Outcome

Hypothesis 3 predicted the outcome differences between regulated and trauma groups for Private, Exposed, and Threatened self. Hypothesis 3 was only supported for the Exposed and Threatened self (χ^2^ = 16.44, *p* < 0.01, χ^2^ = 25.58, *p* < 0.001, respectively). There were no significant differences between regulated and trauma groups for the Private self (χ^2^ = 5.95, *p* > 0.05).

There were, however, significant differences between individual attachment classifications for all three settings. The results are shown in Table 5, Table 6 and Table 7. The shame outcome has the potential to repair biological dysregulation [11]. Differences in outcomes for each attachment group showed how outcome patterns were distributed in each setting. As shown in Table 5, Table 6 and Table 7, there were significant differences among individual attachment groups, including narratives involving the Private self. These results showed that integration or repair were relatively rare outcomes. However, this form of repair characterized the secure groups (F, Ftr) more than the insecure and traumatized groups. The most frequent shame outcomes for all settings were Functional–Self-management and Enduring–Unremedied.

Linear regressions were performed to evaluate the relative contributions of attachment classification and shame story outcomes to shame intensity. Attachment classifications were transformed into a continuous scale of integrated (1) to dysregulated (7). Outcomes were transformed into a continuous scale of integrated (1) to dysregulated (6). Stepwise regression entered the independent variables of attachment, outcome, and the interaction between the independent variables. Participant demographics were not significantly related to shame intensity, so there were no covariates.

The result for the Private self was significant, F(2,57) = 13.35, *p* < 0.001, R^2^ = 0.33. Only attachment predicted shame intensity, β = 0.47, *p* < 0.001. Neither the outcome nor the interaction between attachment and outcome was significant.

The result for the Exposed self was significant, F(2,57) = 26.42, *p* < 0.001, R^2^ = 0.22. Both attachment and outcome contributed to shame intensity for the Exposed self. Attachment was the most significant contributor, β = 0.35, *p* < 0.001. The outcome contribution was less significant, β = 0.19, *p* < 0.05. The interaction between the independent variables did not make a significant contribution.

The result for the Threatened self was significant, F(2,57) = 76.76, *p* < 0.001. Attachment made the most significant contribution, β = 0.49, *p* < 0.001, followed by outcome, β = 0.23, *p* < 0.001. The interaction between independent variables did not make a significant contribution.

## 4. Discussion

The purpose of the current study was to refine our thinking about the relationship between attachment and shame. For several decades, researchers and clinicians have associated shame with insecure attachment, with its origins in infancy. Schore’s [13] model of interpersonal neurobiology explains how mother–child right brain attunement failures foster internalized traumatic shame. A central question to understanding adult shame is how to access the developmental influences of childhood attachment experience. Like only two prior studies [11,34], this study approached this question using the Adult Attachment Projective Picture System. The AAP was designed and validated as a free response assessment for adult attachment that helps bypass left brain cortical control and better accesses the right brain and limbic system processes than the interviews or questionnaires popular in the field [30]. The study’s approach to defining trauma is congruent with Schore’s model. The AAP is more trauma-sensitive than other measures and shows sensitivity to right brain dysregulation [38]. In addition, Schore’s neurological approach to trauma overlaps with George and Solomon’s view of attachment trauma in terms of parental abdication and protective failure [29].

Another unique feature of this study is extending Bowlby’s [23] conceptualization of pathological mourning and trauma assessed by the AAP. The AAP identifies not only Unresolved attachment but also individuals who have Failed Mourning or are Preoccupied with Personal Suffering. Further, the assessment identified a group of secure individuals who continue to be “haunted” by childhood trauma. Adding these trauma groups to the traditional Unresolved group permitted the examination of the classification patterns that comprised regulated versus regulated attachment. Other attachment studies only investigated trauma as an Unresolved attachment.

Finally, a third feature of this study was to evaluate adults’ representations of the shamed self in three settings. One was the Private self depicted by a child alone in her home looking out of a window. In addition, the Window is the least stressful stimulus in the AAP picture set [30]. With nobody else in the picture, feelings of shame are privatized. Shame is also the product of the eye of the beholder [5]. The second setting is the Exposed self, which is depicted by a character sitting alone in the middle of nowhere. This picture activates the stress of being on display, exposing the shamed self in a way that one would want to avoid [5]. The third was the Threatened self depicted by a cornered child with their hands up. This scene is the most stressful stimulus in the AAP set [35] and was included to produce stories of conflict and potential abuse [30]. One of the origins of a pervasive shamed self is thought to be emotional or physical abuse [28].

### 4.1. Shame in Adults with Regulated and Traumatized Attachment

To investigate attachment and shame, we first examined the frequency with which individuals responded to the AAP alone self with elements of shame. Hypothesis 1 predicted differences between regulated and trauma classification groups across all of the shame settings. Reviewing the individual classifications, the results showed significant between-group differences for the Private and Exposed self. As would be expected, there were fewer shame stories related to the Private self in all attachment groups. Interestingly, the differences among individual classification groups for the Threatened self only reached a significance level of a trend. Shame stories ranged from 76 to 100% for the entire sample. This finding suggests that regulated individuals are not protected from shame when threatened. Notably, the highest frequency in the regulated group was for Dismissing attachment, with 87% of their stories told in response to Corner. This proportion was not too far off from the frequency seen in the Unresolved group at 89%. Regulated by deactivating defenses, one might have expected a lower story frequency for Dismissing individuals. These findings may show that deactivating defenses are less effective in minimizing feelings of shame than they might be for managing anger or comfort needs.

These patterns of results shed new light on understanding individuals’ descriptions of shame. Although reticent to use the words shame or embarrassment, individuals are not hesitant to describe shamed behavior or the internalized feelings surrounding shame, e.g., [23]. In short, individuals will not name it, but they can see and describe it.

The other hypotheses pertained only to shame stories. Hypothesis 2 investigated shame intensity. The shame intensity scale was designed to delineate gradations of shame representations ranging from shame in everyday situations to harsh and traumatizing shame. Hypothesis 2 predicted differences in shame intensity between the regulated and traumatized groups. This hypothesis was supported in all shame settings. The patterns for individual attachment classifications provide an in-depth picture. The intensity ratings for the regulated attachment groups—Secure, Dismissing, and Preoccupied—were essentially equivalent. Shame intensity for the Unresolved group was the greatest in all shame settings. This finding is consistent with studies demonstrating the relationship and mental health vulnerability associated with Unresolved attachment, e.g., [35,38].

Especially interesting are shame intensity patterns for the Exposed and Threatened self. The deactivating defenses of Dismissing attachment successfully muted shame for the Exposed self; the intensity ratings were similar to the ratings for the Traumatized Secure. Deactivation appears to break down, however, when the self feels threatened. The stories of individuals in failed mourning showed essentially the same shame intensity as those preoccupied with personal suffering. In sum, these results are consistent with other studies that show that the homeostasis associated with security [30,32] acts as a shame buffer, especially for the Threatened self.

The outcome of shame events has the potential to repair and restore relationships. Hypothesis 3 predicted differences in outcomes between regulated and trauma groups for all settings. This hypothesis was only partially supported. There were outcome differences for only the Exposed and Threatened self-settings. We might think that because the shame stories for the Private self were relatively rare, combined with the Window picture not being very stressful, the outcome may not be important to the conversation about internalized shame. The examination of the individual attachment groups shows otherwise, however. Consistent with the Solomon and George study [11], integrated repair predominated for the Secure group. The association of repair with Traumatized Secure was somewhat surprising because this group experiences lingering trauma. This result suggests that the Traumatized secure group has more in common with individuals with regulated attachment than those in incomplete mourning.

In terms of overall outcomes, these authors expressed surprise that repair was not more common in their study. The findings of the current study were the same. Outcomes tended to be more functional than reparative. Many stories bypassed repair and described repair attempts contaminated by scolding, explanations for why the child was shamed, ways to make restitutions, or the child having to manage their shame on their own without parental guidance or help. In response to typical shame contexts, parents and others may bypass repair to emphasize the socialization point of “Do not do this again”. It was surprising, however, how many stories ended without remedy. Enduring shame was observed in the stories across all groups and shame settings. Some stories showed, for example, that even if a parent apologized, for example, the apology did not assuage mistrust and the child continued to be sad or upset. If the bullying friend tried to repair the incident, the teen would still be confused and angry.

### 4.2. The Relative Contribution of Attachment Patterns and Shame Outcomes to Shame Intensity

The linear regression explored if and how attachment or outcomes contributed more to shame intensity. There was no interaction between these sources. Attachment was the sole predictor for the Private self. The attachment also contributed the most to the intensity of shame for the Exposed and Threatened self. However, outcomes were also significant. These findings suggest that an individual’s attachment pattern (i.e., relative security) may provide a representational frame that sets the stage for how individuals experience shame and respond to the outcome. Individuals with Secure, Dismissing, and Traumatized Secures may be able to accept functional or even enduring outcomes because of their enduring confidence that their attachment figures will not abandon them. Individuals with other forms of insecurity would not have the same confidence. Those with Preoccupied attachments (E) are chronically confused about attachment figure availability and responsiveness. Individuals in the pathological mourning groups struggle with reminders of attachment figure failures (i.e., Dstr Etr, U). These individuals would be unable to trust that they will obtain the care they need; their shame is internalized as a component of the core self.

Although the study yielded important new findings, there were also limitations. The study used archival data from two samples. The attachment distributions were similar. However, the number of secure adults in both samples was smaller than expected in community samples [39]. This distribution of these participants reflects how they were recruited for these studies. The goal of the mother study was emotional development, which tends to appeal to mothers of insecure children [32]. The adult sample aimed to examine stress, similarly appealing strongly to stressed individuals.

Another limitation is that the use of the AAP in shame research is relatively new. The shame and outcome categories assessed in this study were created from the clinical shame literature. The intensity rating was developed to provide a continuum view of the kind of shame responses described by Solomon and George [11]. More research is required to validate the use of the AAP in examining shame and the rubric used in the current study.

## 5. Conclusions

The current study provides a unique view of the association between attachment and shame. Using the AAP offers a deeper understanding of the role of attachment and trauma than in the field to date. It aligns the concept of attachment trauma with an interpersonal neurobiological view of shame. This study demonstrates the AAP as a feasible measure to assess adult attachment representations and internalized shame originating in attachment experiences. Although validated to assess early attachment experiences, life experiences continue to contribute to self-representation. The AAP demonstrates “where” individuals are now in their self-view and relationships. This view especially provides clinicians with informative guidelines about a client’s past and present to introduce ideas regarding shame to their clients and expectations for therapy.

The study adds to the attachment literature by poising shame as a powerful contributor to emotional development and the quality of attachment relationships. Bowlby’s second volume [41] describes what evolutionary biologists consider “natural cues” of danger. Bowlby argued that separation from the parent is a natural danger cue as important as fear or pain. The current study suggests that shame, which signifies a threat to attachment, is a potential separation. It should also be viewed as a natural cue of danger. Like the other cues Bowlby describes, shame in and of itself is not necessarily pernicious. This study clarifies what it means to experience normal rejection in socialization settings, showing that it compromises the attachment relationship but is not dangerous. Enduring and unrepaired shame beyond essential socialization is abandonment, not rejection. Feelings of abandonment and correlative isolation create a core enduring representation of self as worthless and undeserving. In normal situations, shame communicates a clear message that misdoing is unacceptable. In extreme situations, the cost of shame is mis-attunement that dysregulates neurophysiological and emotional development and traumatizes attachment.

## Figures and Tables

**Table 1 brainsci-15-00415-t001:** Shame categories.

Shame Categories	Description
Names shame	Evaluations of the self character using the words shame, ashamed, embarrassed, shy, or silly.
Behavior	Descriptions of (1) shame postures (e.g., the character hides their face so one will not see their feelings, the character is curled up, the teen is sad and slumped over because they lost the swim meet) or (2) behavioral withdrawal (e.g., wanting to get away and not talk to anyone; being scolded and going to their bedroom to hide).
Core shame	Self-deprecating internalized representations of being unworthy or unlovable (e.g., the person knows they are the problem or at fault and wants to commit suicide).
Shame defense	(1) Derealization or dissociative–depersonalized imagery (e.g., the character looks away and pretends they cannot be seen; they want to drop to the other side of the earth; they kill the real self and create a new self). (2) The character attempts to appease the shamer (e.g., begs and desperately promises never to misbehave again).
Shame contexts	Situations involving (1) normative (e.g., the parent punishes the child’s transgression or lectures them about correct behavior) or (2) traumatizing dysregulating shame (e.g., the character is in jail or a mental institution, the parent screams, hits, or threatens the misbehaving child; the character feel trapped, overwhelmed, depressed, or helpless; parent–child role reversal).

**Table 2 brainsci-15-00415-t002:** Shame intensity rating scale.

Rating	Description
0 No shame	The story does not describe shame contexts or emotional correlates (see Table 1).
1 Adaptive	The shame context is adaptive, i.e., normal, or would be expected for that situation. There is no evidence of dysregulating shame (e.g., fear, helplessness, isolation). The character may attempt to appease the shamer (e.g., the parent scolds the child for being mean to their sibling).
2 Adaptive with minimal dysregulation	The shame context is adaptive, i.e., normal, or would be expected in a situation, with one or two indicators of dysregulating shame (e.g., the character feels isolated because he is made to feel left out).
3 Manageable dysregulation	The shame context is adaptive but moderately dysregulating, i.e., normal, or would be expected in a situation (e.g., other kids are making fun of the child at a birthday party and he feels overwhelmed, overstimulated, and helpless).
4 Intense	The narrative evidences a moderate amount of traumatizing dysregulating shame, including self-deprecating statements (e.g., the child feels coerced and unsafe by his parent’s demands and feels helpless and tells them to “please back off” because he feels threatened).
5 Harsh	Harsh or traumatic contexts and corollary emotions define the narrative. Stories that evidence derealization or dissociative imagery or parent–child role reversal are rated a 6 (e.g., her partner dumped her and she’s homeless and in despair; she has her face buried in her arms; she meets up with some people and does drugs; feels isolated and stares out at the solar system).
6 Terrifying	The narrative is defined by contexts of frightening threat, abuse, and attack combined with evidence of helplessness, isolation, and in some stories derealization/dissociation (e.g., the father is abusive, and lashes out, yelling and screaming at the kid; the mother berates him, “Why do you start fights with your Dad? You know the way his is” the kid wishes his father would die. He shuts down and tries to think of something else to replace what is happening to them).

**Table 3 brainsci-15-00415-t003:** Attachment classification and frequency of shame stories for Private, Exposed, and Threatened self settings.

Attachment Classification	Frequency Private Exposed Threatened	%	χ^2^
F (*n* = 33)	2 13 25	6% 39% 76%	Private: 18.09 ** Exposed: 47.37 *** Threatened: 11.81 ^+^
Ds (*n* = 31)	2 18 27	6% 58% 87%	
E (*n* = 18)	4 10 15	22% 55% 75%	
Ftr (*n* = 36)	8 27 34	22% 75% 94%	
Dstr (*n* = 53)	14 46 50	26% 33% 94%	
Etr (*n* = 18)	6 17 18	33% 94% 100%	
U (*n* = 56)	21 52 50	38% 68% 89%	

*** *p* < 0.001, ** *p* < 0.01, ^+^ *p* < 0.10.

**Table 4 brainsci-15-00415-t004:** Attachment classification and mean shame intensity ratings for the Private, Exposed, and Threatened self.

Attachment Classification	Mean Intensity Rating (SD)	Minimum–Maximum Scores	Kruskal–Wallace Statistic	Between-Group Differences
F Private *n* = 2 Exposed *n* = 13 Threatened *n* = 23	1.00 (0.00) 1.69 (0.95) 1.74 (0.75)	1–1 1–4 1–3	Private: 20.41 **	Private: F, Ds, E < Ftr, Dstr, Etr < U
Ds Private *n* = 2 Exposed *n* = 18 Threatened *n* = 26	1.50 (0.71) 1.83 (0.86) 1.81 (0.94)	1–2 1–4 1–4	Exposed: 43.05 ***	Exposed: F, Ds, E < Ftr, Dstr < Etr, U
E Private *n* = 4 Exposed *n* = 10 Threatened *n* = 12	1.25 (0.50) 1.20 (0.42) 1.92 (0.67)	1–2 1–2 1–3	Threatened: 90.84 ***	Threatened: F, Ds, E < Ftr < Dstr, Etr < U
Ftr Private *n* = 8 Private *n* = 27 Exposed *n* = 32	2.25 (1.49) 2.59 (1.28) 2.69 (1.38)	1–5 1–5 1–5		
Dstr Private *n* = 14 Exposed *n* = 46 Threatened *n* = 51	2.25 (1.45) 2.83 (1.51) 4.04 (1.52)	1–5 1–6 1–6		
Etr Private *n* = 6 Exposed *n* = 17 Threatened *n* = 18	3.83 (1.47) 3.35 (1.65) 4.22 (0.94)	2–5 1–6 1–5		
U Private *n* = 21 Exposed *n* = 52 Threatened *n* = 53	4.00 (1.73) 3.61 (1.57) 4.30 (1.40)	1–6 1–6 1–6		

** *p* < 0.01, *** *p* < 0.001.

**Table 5 brainsci-15-00415-t005:** Attachment classification and shame outcomes for the Private self.

Attachment Classification	F *n* (%)	Ds *n* (%)	E *n* (%)	Ftr *n* (%)	Dstr *n* (%)	Etr *n* (%)	U *n* (%)
Repair	1 (50%)	0	0	5 (62%)	0	0	1 (45)
Restore	0	0	0	0	0	0	0
Functional	0	0	0	2 (25%)	6 (43%)	0	1 (4%)
Self-manage	1 (50%)	1 (50%)	2 (40%)	1 (13%)	5 (36%)	1 (17%)	5 (23%)
Enduring	0	1 (50%)	3 (60%)	0	3 (21%)	5 (83%	4 (18%)
Unremedied	0	0	0	0	0	0	11 (50%)
Total	2	2	5	8	14	6	22

χ^2^ = 69.26, *p* < 0.001.

**Table 6 brainsci-15-00415-t006:** Attachment classification and shame outcomes for the Exposed self.

Attachment Classification	F *n* (%)	Ds *n* (%)	E *n* (%)	Ftr *n* (%)	Dstr *n* (%)	Etr *n* (%)	U *n* (%)
Repair	6 (46%)	0	0	12 (44%)	0	0	1 (1%)
Restore	2 (15%)	3 (17%)	0	3 (11%)	2 (4%)	0	0
Functional	1 (8%)	6 (33%)	4 (37%)	4 (15%)	9 (20%)	4 (24%)	6 (12%)
Self-manage	3 (23%)	6 (33%)	1 (9%)	7 (26%)	13 (28%)	0	4 (8%)
Enduring	1 (8%)	3 (17%)	6 (54%)	1 (4%)	22 (48%)	13 (76%)	17 (33%)
Unremedied	0	0	0	0	0	0	24 (46%)
Total	13	18	11	27	46	17	52

χ^2^ = 176.64, *p* < 0.001.

**Table 7 brainsci-15-00415-t007:** Attachment classification and shame outcomes for the Threatened self.

Attachment Classification	F *n* (%)	Ds *n* (%)	E *n* (%)	Ftr *n* (%)	Dstr *n* (%)	Etr *n* (%)	U *n* (%)
Repair	12 (52%)	0	0	12 (38%)	1 (2%)	0	0
Restore	1 (4%)	4 (15%)	0	1 (3%)	5 (10%)	1 (6%)	0
Functional	2 (9%)	9 (35%)	0	7 (22%)	7 (14%)	1 (6%)	3 (6%)
Self-manage	2 (9%)	8 (31%)	2 (18%)	4 (12%)	7 (14%)	0	2 (3%)
Enduring	6 (26%)	5 (19%)	9 (82%)	8 (25%)	30 (60%)	15 (82%)	20 (38%)
Unremedied	0	0	0	0	0	1 (6%)	28 (53%)
Total	23	26	11	32	50	18	53

χ^2^ = 207.46, *p* < 0.001.

## Data Availability

The original contributions presented in this study are included in the article. Further inquiries can be directed to the corresponding author.

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
