# Peer review of "Attachment, Shame, and Trauma"

_brainsci, 2025, doi:10.3390/brainsci15040415_

Round 1

Reviewer 1 Report

Comments and Suggestions for Authors

Thank you for an engaging and interesting article. I think this is an original contribution to knowledge in this area building on a limited range of previous studies.  The research seems to have been carried out appropriately and the results presented clearly.  The implications of the research could be a topic for further study.

I am interested in your choice of authors to frame your definitions in the introduction and wonder if some additional information as to who the authors are and why they have been chosen would be useful - this is inconsistent through the article.   Given the breadth of writing on shame, choosing to use something with non-inclusive language without indicating that this could be an issue is an interesting choice. Thus I would suggest changing reference 2, you go on to cite some other well respected authors in that paragraph.

Section 2.1, it would be good to have further details around the choice of these two particular cohorts to study.

Line 47 typo on evidence.

Line 127 doesn't quite make sense.

Typo l159

Lines 192-3 this is very dependent on age, perhaps some discussion on what is meant by child in this discussion would be useful.

Line 202 typo on grief

L271 assume that should say elaborating

Author Response

Reviewer 1

Response to reviewers 1

Cited authors: The scholars selected to frame this paper include important biologists (Schore, Tomkins), developmental psychologists who have done the most work in shame (Michael Lewis), and noted shame scholars in the clinical psychology literature. I propose the lack of consistency noted is not a lack of consistency in the paper but a lack of consistency in the shame literature. As an attachment theorist, I begin with biology (Bowlby's approach combined with neurophysiology), which naturally leads to developmental ideas. One of the main points of this paper is to demonstrate empirically, especially to clinical psychologists, that not all shame is traumatizing. This group of scholars has been sloppy, in the authors' view, in their use of attachment and assumptions about trauma and shame. I am sorry that I don't understand what you mean about using non-inclusive language, so clarification would be helpful.

Reference 2 is from a book describing Tomkins' work, which is important and central to my thinking. I would prefer to keep this reference because here I begin with Tomkins. If that reference is confusing or inconsistent, it would not detract from the paper to delete that whole paragraph and quote and go from "loss of face" to sentence 41.

2.1 about the samples – I included a bit more information about the samples.

“Data for the combined two cross-sectional attachment studies that were being conducted simultaneously in the same lab to investigate correlates and sequelae of attachment." These studies were conducted simultaneously, and recruitment was going on simultaneously but with separate goals. In many studies, there are too few participants for statistical power. Since these were simultaneous groups that showed no differences in attachment or gender, they were combined to provide a larger sample for power and a more diverse group of people in the community.

Line 47, typo corrected. And all of the other typos in the manuscript.

Thank you for pointing out the confusion in Line 127. Building the case for mourning was complicated. The wording now is, "Following this thinking, the only way to repair shame is to mourn.”

L 271, elaborating, yes. Thank you for catching that error.

Reviewer 2 Report

Comments and Suggestions for Authors

Dear Author,

This manuscript presents a well-conceived and methodologically rigorous examination of the relationship between attachment and shame. The study is grounded in a strong theoretical framework that effectively integrates attachment theory with interpersonal neurobiology, providing a comprehensive perspective on how early attachment experiences shape the internalization and regulation of shame. A notable strength is the use of the Adult Attachment Projective System (AAP), which allows for a more implicit and dynamic assessment of attachment representations than self-report measures. This approach enhances the ecological validity of the study and contributes to the growing body of research supporting projective methods in attachment research.

The results significantly contribute to the field, particularly in clarifying the relationship between unresolved attachment and increased shame intensity. The distinction between regulated and dysregulated attachment patterns is particularly insightful, highlighting how individuals process and respond to shame-related experiences. The discussion section effectively contextualizes these findings within existing theoretical models while emphasizing their clinical relevance. The implications for psychotherapy are particularly noteworthy, highlighting the importance of addressing shame within attachment-based interventions.

The methodological rigor of the study further strengthens its contribution. The sample size is adequate and the statistical analyses are well conducted. Using regression models to assess the predictive power of attachment classification on shame intensity adds robustness to the findings. In addition, the procedures for coding shame narratives are well justified and demonstrate strong inter-rater reliability.

Overall, this manuscript represents a valuable contribution to the literature and is suitable for publication in its current form. The study is well articulated, methodologically sound, and provides significant findings that will be of interest to researchers and clinicians alike. Congratulations on this excellent paper!

Author Response

Reviewer 2.

Comments:

Dear Author,

This manuscript presents a well-conceived and methodologically rigorous examination of the relationship between attachment and shame. The study is grounded in a strong theoretical framework that effectively integrates attachment theory with interpersonal neurobiology, providing a comprehensive perspective on how early attachment experiences shape the internalization and regulation of shame. A notable strength is the use of the Adult Attachment Projective System (AAP), which allows for a more implicit and dynamic assessment of attachment representations than self-report measures. This approach enhances the ecological validity of the study and contributes to the growing body of research supporting projective methods in attachment research.

The results significantly contribute to the field, particularly in clarifying the relationship between unresolved attachment and increased shame intensity. The distinction between regulated and dysregulated attachment patterns is particularly insightful, highlighting how individuals process and respond to shame-related experiences. The discussion section effectively contextualizes these findings within existing theoretical models while emphasizing their clinical relevance. The implications for psychotherapy are particularly noteworthy, highlighting the importance of addressing shame within attachment-based interventions.

The methodological rigor of the study further strengthens its contribution. The sample size is adequate and the statistical analyses are well conducted. Using regression models to assess the predictive power of attachment classification on shame intensity adds robustness to the findings. In addition, the procedures for coding shame narratives are well justified and demonstrate strong inter-rater reliability.

Overall, this manuscript represents a valuable contribution to the literature and is suitable for publication in its current form. The study is well articulated, methodologically sound, and provides significant findings that will be of interest to researchers and clinicians alike. Congratulations on this excellent paper!

Response: Thank you for your appreciation of this study.

Reviewer 3 Report

Comments and Suggestions for Authors

While the manuscript might be suitable for a journal in social psychology, general psychology, and other areas like developmental psychology, it does not seem appropriate to the stated scope of Brain Sciences

This is because, in spite of occasional references to neurobiology, they are almost always inserted without anything that I would count as neurobiological explanations. Even the section titled "The Biology of Shame" basically does not talk about biology.

The author does cite some interesting work on the neurobiology of shame, but again without really talking about anything specifically neurobiological. 

Some helpful sources might include some of Jay Schulkin's books. Though his books are hit or miss (i.e., quite a few of them are not worthwhile), some are decent and accessible, offering intriguing neurobiological accounts of shame.

A few other helpful sources might be:

  • Piretti et al. (2023) "The Neural Signatures of Shame, Embarrassment, and Guilt: A Voxel-Based Meta-Analysis on Functional Neuroimaging Studies"
  • Moll et al. (2005) “The Neural Basis of Human Moral Cognition.”

Another problem is that many of the discussions are quite bloated, and this is from the standpoint of a reviewer who normally appreciates philosophical and theoretical discussions.

For example, everything in sections 1–1.3 could be vastly distilled without detracting from the substance. Likewise, the discussion of the measures that the author deploys in section 2 could be significantly trimmed.

 I would also suggest dividing section 4 into at least one additional subsection to increase readability.

The bloated quality of the text (and I am really not trying to be mean) makes it difficult to give more specific criticism — at least not without putting hours and hours of additional work into the review. 

None of this is to deny that the manuscript has merit. 

But I would suggest that if the author wishes to submit to a more appropriate journal, such as a social psychology journal, efforts should be put into significantly streamlining the paper.

Finally, and more favorably, in clicking the box indicating that "The English could be improved to more clearly express the research," I do not mean to suggest that the writing is bad. It is instead just that there is virtually no paper (my own work included) where the language could not be improved.

Comments on the Quality of English Language

English is fine. My indication that the English could be improved is a statement I would make for virtually any manuscript.

Author Response

Reviewer 3.

I agree that there could be a more elaborate biological description. Even including more from Schore’s work – who has done the best work on shame in the literature – would increase the length of the manuscript. This is not to say that other models are unwelcome, but Schore’s work is the only comprehensive neuro-attachment thinking that crosses both fields. Including the details of this thinking, however, would make the paper longer than acceptable at the expense of desribing the unique fearture of the study. Similarly, other work discussing the neurological underpinnings of attachment would also include the special editor's studies on attachment, but they do not address shame. The paper aimed to integrate the two fields – neuro and attachment theory. While Skulkin's books and others listed are interesting from the neuro perspective, integrating these approaches with attachment would require a chapter and is beyond the goal of this special issue.

I respectfully disagree that the discussion presented in the first section are bloated. There are so few developmental studies, much less attachment studies about shame, that readers appreciate knowing the details without needing to find the studies themselves. Additionally, the studies that are described serve as the foundation of the current study and provide background for the coding system described in the Method section. Another feature of that section addresses clinical psychologists who equate attachment insecurity with shame, which is frankly irresponsible because they neither understand nor describe the nuances of attachment patterns. One paper goal is to create discourse precision.

With due respect to the reviewer, it is essential to clarify that attachment, as addressed in this study, is not a social psychological phenomenon. This paper is not better suited to a social psychology journal. Attachment originated and is, in this paper, a developmental biological survival mechanism. Unfortunately, social psychology usurped attachment theory, and now, years later, the differences between the two streams are obscured. It is not surprising that readers are confused. Developmentalists examine early parent-infant relationships' lifelong psychological, relationship, emotional, and psychiatric effects and how attachment patterns are transmitted intergenerationally. The measure used in this study has been validated following this line of thinking, as are other developmental measures such as the Adult Attachment Interview for adolescents and adults, variations of the Strange Situation for children 12 months to age 7, and symbolic play assessment used to assess children ages 4 ½ to 12.

The history of attachment style, the term created by social psychologists, has been obscured in part because it is convenient to to use their questionnaires. It's purely empirical. Romantic attachment (a synonymous term used in social psychology) is not synonymous with the original Bowlby-Ainsworth model. A detailed account of the early synthesis of their view of attachment with personality theory is described in a paper in the British Journal of Medical Psychology, https://doi.org/10.1348/000711299159998. The origins of attachment style were in explaining adult loneliness; concurrent validity of the groups assessed using their questionnaires was established in part using measures of marital satisfaction, including sexual satisfaction. There is no established empirical relationship between the two approaches, including the many longitudinal studies that followed infants throughout the lifespan, including when infants grew up to have their own children (see Sroufe's University of Minnesota studies).

Section 4 is now separated into two sections. Section 4.2 focuses on the regression analysis, which demonstrates the separate contributions of attachment patterns (including incomplete mourning) and outcomes to the intensity of individuals' pain internalized from being shamed.

Reviewer 4 Report

Comments and Suggestions for Authors

The topic of attachment, trauma, and shame is crucial, as it provides valuable insights into assessing attachment and shame for effective goal-setting in therapy. However, some sentences are difficult to follow, and several citations need correction. I recommend a thorough revision to enhance clarity, refine certain passages, and ensure accurate citations.

My comments-

Line 8; “The purpose of this study was to refine our thinking about the relation between attachment and shame”. This sentence can be changed, or at least change “refine our thinking”.

Line 21;  This sentence is not clear. “There was a significant increase was observed in the traumatized groups, with the Unresolved group showing the highest ratings.”

Line 23-24; “secure regulated insecure”, what does it mean?

Line 26; “more significant” should be changed to “significant”

Line 35; “foolish behavior” could be replaced with other word. Eg, self-loathing or absurd.

Line 36; Author cited ref for 1, whereas, in the reference list number 1 is “American Heritage Dictionary of the English Language.”

Line 38, line 39, line 40; “……..” should be removed.

Few sentence should be fixed, like “Heart of man”, “He feels”

Line 116; Author cited Solman and George “17, and 18” whereas, only ref number 17 is of Solman and George in the reference list.

Line 125; Bowlby (19) is wrongly cited. It should be “18” according to the reference list.

Line 194; Kaufman cited number “22, 25”. Only citation number 22 is of Kaufman, not the 25.

Line 198; Tomkins (2) is wrongly cited. In the reference list Tomkins’s article is at number 3.

Line 209-212; sentence is too big to understand. Please rewrite it.

Line 217; this sentence can be improved.

I think overall introduction can be shorten.

Line 277; The sentence start with “it was explain earlier..” It should be cited that author meant it was explain earlier.

Line 284; Author should check if meant for “lonely” or “alone”.

Author Response

Reviewer 4

 Response to reviewer 4

Thank you for your recommendations and requests for clarification.

L8: The sentenced recommended for change now reads: “The purpose of this study was to advance our understanding of the relation between attachment and shame.”

L21: The redundant verb “was” is removed from the sentence. This error missed proof reading. Thank you.

L23-24: The question indicates that the concept of “regulated insecure” may be unfamiliar to readers even though it is standard in the field of attachment. The following clarification of which group patterns are included in this terminology is added: regulated insecure (Dismissing, Preoccupied).

L26: Removing “more” from the sentence doesn’t describe the finding. Therefore, the phrase about prediction is changed to: “Regression analysis demonstrated that attachment classification was a greater predictor of shame intensity than outcomes.”

L35, L36: This statement as written is a quote from the dictionary. Rather, than use the quote verbatim, the statement is paraphrased to read: “Shame is a painful feeling of humiliation or distress caused by consciousness awareness of wrong or absurd behavior [1].” The reference question should no longer be an issue since the wording is changed.

L38-40: The quote is edited. One cannot add capitals where capitals are not in the original quotation, so “heart of man” remains unchanged.

L116: The double reference [17,18] is corrected to list just [17].

L 125 etc. These are apparently an EndNote errors and it was a proofreading error to trust the final EndNote. Thank you for catching this. The reference numbers are corrected.

L209-212: The sentence was shortened to read: “Pathological mourning is associated with a range of maladaptive defenses that overlap with shame as described in the clinical literature.”

L217: The sentence now reads: “The AAP is an empirically validated free response measure of adults’ mental representations of childhood attachment experiences.”

Shorten: Thank you for this recommendation. The discussion in the introduction is reduced to be more concise and reader friendly. The descriptions of the two shame studies that set the stage for the current study are reduced to the bare essentials.

L277: The explained earlier is not needed. That phrase is deleted. This was the author speaking directly to the clinicians and does not add to the presentation about the study.

L 283: The correct term is “alone” (Bowlby, 1969)

Round 2

Reviewer 3 Report

Comments and Suggestions for Authors

It’s not so much that I was arguing shame is an inappropriate topic for a journal focused on brain sciences, but rather that the paper did not engage sufficiently with biological matters to warrant inclusion in a journal devoted to brain science.

I don’t have a vested interest in whether the editors choose to include the paper, so long as they feel it aligns with the journal’s scope.

If they do want to include the piece, then one suggestion for the author: since the author believes I misunderstood the work in suggesting it might be more appropriate for a social psychology journal—and given the author mentions there are historical misunderstandings on this point—it might be worthwhile to include a few paragraphs that clearly explain why the work is biologically oriented (and better still, neurobiologically oriented). My philosophy with reviews is that even when I think reviewers have misunderstood my own work, I try to address the misunderstanding on the assumption that others might similarly err. This is, in fact, a point the author themselves makes, so why not clarify a little more?

Regarding the section I previously called “bloated,” another way to put it is that it was unwieldy. It was a bit of a slog getting through that part and seeing where the author was headed—and this comes from a reviewer accustomed to highly theoretical discussions (while I do work with both quantitative and qualitative data, most of my research is more theoretical). Although the author has trimmed it somewhat, I still suggest giving the manuscript another careful edit with an eye to enhancing overall clarity.

Author Response

Response to reviewer 3 16 April 2025

Reviewer 3.

Comments: It’s not so much that I was arguing shame is an inappropriate topic for a journal focused on brain sciences, but rather that the paper did not engage sufficiently with biological matters to warrant inclusion in a journal devoted to brain science.

I don’t have a vested interest in whether the editors choose to include the paper, so long as they feel it aligns with the journal’s scope.

If they do want to include the piece, then one suggestion for the author: since the author believes I misunderstood the work in suggesting it might be more appropriate for a social psychology journal—and given the author mentions there are historical misunderstandings on this point—it might be worthwhile to include a few paragraphs that clearly explain why the work is biologically oriented (and better still, neurobiologically oriented). My philosophy with reviews is that even when I think reviewers have misunderstood my own work, I try to address the misunderstanding on the assumption that others might similarly err. This is, in fact, a point the author themselves makes, so why not clarify a little more?

Regarding the section I previously called “bloated,” another way to put it is that it was unwieldy. It was a bit of a slog getting through that part and seeing where the author was headed—and this comes from a reviewer accustomed to highly theoretical discussions (while I do work with both quantitative and qualitative data, most of my research is more theoretical). Although the author has trimmed it somewhat, I still suggest giving the manuscript another careful edit with an eye to enhancing overall clarity.

Response:

Thank you for your suggestion to include a discussion of the neurobiological foundation of attachment. It is easy to assume as an attachment theorist and researcher for 50 years that readers would know the literature and make the connection without evidence. This was an oversight. A section at the beginning of the paper is added on neurobiology. There would be so much to review, so this section lets the reader know that a full review of this literature is beyond the scope of the paper. 

As for the length, there is a lot of new material in the introduction that is contrary to clinical approaches to shame and elaborates in unique ways on attachment theory view of trauma. Like unfamiliarity with the neurobiology of shame, the reader is unlike to be familiar with these points as well. The introduction is shortened. The reviewer will find that – except with the new section on the neurobiology of attachment – the introduction including the hypotheses is reduced by 711 words.
